# Effects of a Physical Education Program on Physical Activity and Emotional Well-Being among Primary School Children

**DOI:** 10.3390/ijerph18147536

**Published:** 2021-07-15

**Authors:** Irina Kliziene, Ginas Cizauskas, Saule Sipaviciene, Roma Aleksandraviciene, Kristina Zaicenkoviene

**Affiliations:** 1Educational Research Group, Institute of Social Science and Humanity, Kaunas University of Technology, Kaunas 44249, Lithuania; 2Department of Mechanical Engineering, Faculty of Mechanical Engineering and Design, Kaunas University of Technology, Kaunas 51424, Lithuania; ginas.cizauskas@ktu.lt; 3Department of Applied Biology and Rehabilitation, Lithuanian Sports University, Kaunas 44221, Lithuania; saule.sipaviciene@lsu.lt; 4Department of Coaching Science, Lithuanian Sports University, Kaunas 44221, Lithuania or romanellagrande@gmail.com (R.A.); Kristina.zaicenkoviene@lsu.lt (K.Z.); 5Sports Centre, Vytautas Magnus University, Kaunas 51211, Lithuania

**Keywords:** physical educational program, physical activity, anxiety, primary education

## Abstract

(1) Background: It has been identified that schools that adopt at least two hours a week of physical education and plan specific contents and activities can achieve development goals related to physical level, such as promoting health, well-being, and healthy lifestyles, on a personal level, including bodily awareness and confidence in physical skills, as well as a general sense of well-being, greater security and self-esteem, sense of responsibility, patience, courage, and mental balance. The purpose of this study was to establish the effect of physical education programs on the physical activity and emotional well-being of primary school children. (2) Methods: The experimental group comprised 45 girls and 44 boys aged 6–7 years (First Grade) and 48 girls and 46 boys aged 8–9 years (Second Grade), while the control group comprised 43 girls and 46 boys aged 6–7 years (First Grade) and 47 girls and 45 boys aged 8–9 years (Second Grade). All children attended the same school. The Children’s Physical Activity Questionnaire was used, which is based on the Children’s Leisure Activities Study Survey questionnaire, which includes activities specific to young children (e.g., “playing in a playhouse”). Emotional well-being status was explored by estimating three main dimensions: somatic anxiety, personality anxiety, and social anxiety. The Revised Children’s Manifest Anxiety Scale (RCMAS) was used. (3) Results: When analysing the pre-test results of physical activity of the 6–7- and 8–9-year-old children, it turned out that both the First Grade (92.15 MET, min/week) and Second Grade (97.50 MET, min/week) participants in the experimental group were physically active during physical education lessons. When exploring the results of somatic anxiety in EG (4.95 ± 1.10 points), both before and after the experiment, we established that somatic anxiety in EG was 4.55 ± 1.00 points after the intervention program, demonstrating lower levels of depression, seclusion, somatic complaints, aggression, and delinquent behaviours (F = 4.785, *p* < 0.05, P = 0.540). (4) Conclusions: We established that the properly constructed and purposefully applied eight-month physical education program had positive effects on the physical activity and emotional well-being of primary school children (6–7 and 8–9 years) in three main dimensions: somatic anxiety, personality anxiety, and social anxiety. Our findings suggest that the eight-month physical education program intervention was effective at increasing levels of physical activity. Changes in these activities may require more intensive behavioural interventions with children or upstream interventions at the family and societal levels, as well as at the school environment level. These findings have relevance for researchers, policy makers, public health practitioners, and doctors who are involved in health promotion, policy making, and commissioning services.

## 1. Introduction

Teaching in physical education has evolved rapidly over the last 50 years, with a spectrum of teaching styles [1], teaching models [2], curricular models [3], instruction models [4], current pedagogical models [5,6], and physical educational programs [7]. As schools provide benefits other than academic and conceptual skills at present, we can determine new ways to meet different goals through a variety of methodologies assessing contents from a multidisciplinary perspective. Education regarding these skills should also be engaged following a non-traditional methodology in order to overcome the lack of resources in traditional approaches and for teachers to meet their required goals [8].

Schools are considered an important setting to influence the physical activity of children, given the amount of time spent at school and the potential for schools to reach large numbers of children. Schools may be a barrier for interventions to promote physical activity (PA). Children are required to sit quietly for the majority of the day in order to receive academic lessons. A typical school day is represented by approximately 6 h, which may be extended by 30 min or longer if the child is provided motorized transportation and does not actively commute to and from school. Donnelly et al. [9] found that teachers who modelled PA by active participation in physical activity across the curriculum (i.e., promoted 90 min/week of moderate to vigorous physically active academic lessons; 3.0 to 6.0 METs, ∼10 min each) had greater SOFIT (a Likert scale from one to five, anchored with lying down for one and very active for five) scores shown by their students, compared to primary students with teachers using a lower level of modelling. Some studies have proposed the use of prediction models of METs for children, including accelerometer data. In such models, the slope and intercept of ambulatory activities (e.g., walking and running) differ from those of non-ambulatory activities, such as ball-tossing, aerobic dance, and playing with blocks [10,11]. Wood and Hall [12] found that children aged 8–9 years engaged in significantly higher moderate to vigorous physical activities during team games (e.g., football), compared to movement activities in PE lessons (e.g., dance).

It has been identified that schools which adopt two hours a week of PE and plan specific contents and activities to achieve development goals at the physical level can promote health, well-being, and healthy lifestyles on a personal level, including bodily awareness and confidence in one’s physical skills, as well as a general sense of well-being, greater security and self-esteem, sense of responsibility, patience, courage, and mental balance at the social level, including integration within society, a sense of solidarity, social interactions, team spirit, fair play, and respect for rules and for others, as well as wider human and environmental values [13,14]. Physical activity programs have been identified as potential strategies for improving social and emotional well-being in at-risk youth [15]. Emotional well-being permeates all aspects of the experience of children and has emerged as an essential element of mental health and reduction of anxiety, as well as a core component of health in general. Schools have a strong effect on children’s emotional development, and as they are an ideal environment to foster children’s emotional learning and well-being, failing to optimize the opportunity to do so could impact communities in negative ways [16,17]. Physical activity and exercise have positive effects on mood and anxiety, and a great number of studies have described the associations between physical activity and general well-being, mood, and anxiety [18]. Physical inactivity may also be associated with the development of mental disorders: some clinical and epidemiological studies have shown associations between physical activity and symptoms of depression and anxiety in cross-sectional and prospective longitudinal studies [19]. Low physical activity levels have also been associated with an increased prevalence of anxiety [20]. Levels of physical activity lower than those recommended by the World Health Organization are classified as a lack of physical activity or physical inactivity. Current guidelines on physical activity for children and adolescents aged 5–17 years generally recommend at least 60 min daily of moderate- to vigorous-intensity physical activities [21].

Therefore, we formulated the following research hypothesis: The application of a physical education program can have a positive impact on the physical activity and emotional well-being among primary school students.

The purpose of this study was to establish the effect of a physical education program on the physical activity and emotional well-being of primary school children. 

Novelty of the work: For the first time, PE curriculum has been developed for second grade children, a new approach to physical education methodology. For the first time, anxiety is measured between first and second grades. Physical education has been a part of school curriculums for many years, but, due to childhood obesity, focus has increased on the role that schools play in physical activity and monitoring physical fitness [22,23]. 

## 2. Materials and Methods

### 2.1. Participants

The schools utilized in this study were randomly chosen from primary schools in Lithuania. Four schools were chosen from different areas of Lithuania, which are typical of the Lithuanian education system (i.e., the state system), exercising in accordance with the description of primary, basic, and secondary education programs approved by the Lithuanian Minister of Education and Science in 2015. It ought to be noted that these schools structured classes without applying selection criteria; accordingly, it very well may be said that the students in the randomly chosen classes were additionally randomly allocated to the experimental and control groups. A non-probabilistic accurate sample was utilized in the study, where subjects were incorporated relying upon the objectives of the study. 

The time and place of the study, with the consent of the guardians, were settled upon ahead of time with the school administration. This study was approved by the research ethics committee of the Kaunas University of Technology, Institute of Social Science and Humanity (Protocol No V19-1253-03).

The experimental group included 45 young women and 44 young men aged 6–7 years (First Grade) and 48 young women and 46 young men aged 8–9 years (Second Grade). The control group included 43 young women and 46 young men aged 6–7 (First Grade) and 47 young women and 45 young men aged 8–9 years (Second Grade). All children went to a same school.

### 2.2. Instruments

#### 2.2.1. The Evaluation of Physical Activity 

The Children’s Physical Activity Questionnaire [24] was utilized, which is based on the Children’s Leisure Activities Study Survey (CLASS) questionnaire, which includes activities explicit to small children, such as “playing in a playhouse.” The original intent of the proxy-reported CLASS questionnaire for 6–9 year olds was to evaluate the type, recurrence, and intensity of physical activity over a standard week [24].

#### 2.2.2. The Revised Children’s Manifest Anxiety Scale

Enthusiastic well-being status was investigated by estimating three principal dimensions: somatic anxiety, personality anxiety, and social anxiety. The Revised Children’s Manifest Anxiety Scale (RCMAS) contains 37 items with 28 items used to measure anxiety and an additional 9 items that present an index of the child’s level of defensiveness. We were only concerned with the factor analysis of anxiety; along these lines, only those 28 items used to gauge anxiety were utilized. The RCMAS comprises three factors: (1) somatic anxiety, consisting of 12 items; (2) personality anxiety, consisting of 8 items; and (3) social anxiety, consisting of 8 items [25].

The outcomes were estimated as follows: (1) physical anxiety (more than or equal to 6.0 points—high somatic level, from 5.9 to 4.5 points—typical somatic level, and from 4.4 to 1.0 points—low somatic level); (2) personality anxiety (from 2.0 to 2.5 points—low personality anxiety level, from 2.6 to 3.5 points—typical personality anxiety level, and from 3.6 to 4.5 points—high personality anxiety level); and (3) social anxiety (more than or equal to 5.5 points—high social anxiety level, from 5.4 to 4.5 points—typical social anxiety level, and from 4.4 to 3.3 points—low social anxiety level). The Cronbach’s alpha coefficient for the subscales ranged from 0.72 to 0.73.

### 2.3. Procedure

In this study, a pre-/mid-/post-test experimental methodology was utilized, in order to avoid any interruption of educational activities, due to the random selection of children in each group. The experimental group (First and Second Grades) was trialled for eight months. The technique for the physical education program was developed, and a model of educational factors that encourage physical activity for children was constructed.

Likewise, the methodical material for the physical education program [7,24] was prepared. The methodology depended on the dynamic exercise, intense motor skills repetition, differentiation, seating and parking reduction, and the physical activity distribution in the classroom (DIDSFA) model [26,27] (Table 1).

A physical education program was designed in order to advance physical activity to a significant degree, show development skills, and be agreeable. The suggested recurrence of physical education classes was three days out of the week. A typical DIDSFA First Grade model exercise lasted 30 min and had three sections: health fitness activities (10 min), ability fitness activities (15 min), and unwinding, focus, and reflection (5 min). The Second Grade model exercise lasted 45 min and comprised four sections: health fitness activities (20 min), ability fitness activities (20 min), and unwinding, focus, and reflection (5 min). Ten health-related activity units were designed, including aerobic dance, aerobic games, strolling/running, and jump-rope. The movements were developed by changing the intensity, length, and intricacy of the activities.

Although our primary focus was creating cardiovascular stamina, brief activities to develop stomach and chest strength, as well as movement skills, were incorporated. To improve motivation, children self-estimated and recorded their fitness levels from month to month. Four game units which developed ability-related fitness were incorporated (basketball, football, gymnastics, and athletics), and details of healthy lifestyles and unconventional physical activities were introduced. These sports and games had the potential for advancing cardiovascular fitness and speculation in the child’s community (e.g., fun transfers); unwinding, focus, and reflection improving with regular exercise; and valuable impacts for meditation or unwinding, namely through children’s yoga (Table 2).

During the study, physical education activities were taught through physical schooling, by preparing a textbook comprising two interrelated parts: (a) a textbook and (b) children’s notes. The textbooks were filled with logical tasks, self-evaluation, and activities relating to spatial perception and self-improvement. The methodological devices provide strategies for practicing with textbooks. The physical education pack considers a “natural” kind of integration and dynamic learning, building awareness, encouraging sensitivity to nature, and supporting healthy styles of living. The physical education pack takes into consideration a “natural” kind of integration and dynamic learning, building awareness, encouraging sensitivity to nature, and supporting healthy styles of living. The instructor’s manual has a unified structure, which makes it simple to utilize. Its proposals and advice are clear. The advanced version helps educators in their planning and execution activities.

The material seriously assesses intercultural mindfulness and sensitivity. The gender description is balanced; the two personalities highlighted in the textbook support this methodology. Vaquero-Solís et al. found that mixed procedures in their interventions, executed using a new methodology, greatly affected the participants [30]. Once each month, the standard methodology was applied, during which the change from hypothesis to practice was continuous. During the first exercise of the month, the material in the textbook was analysed for the future, and undertakings for the month were presented. The hypothesis was set up during practical sessions. During the hypothetical exercises, the children additionally had the chance to move around, practising the physical tasks given in the textbook. During the last exercise of the month, the tasks introduced in the textbook were performed; the activities of the month were rehashed, recalled, summed up, and assessed; and the assignment of children’s notes were performed. Children from the control group attended unmodified physical education exercises.

### 2.4. Data Analysis 

Graphic statistics are presented for all methodical factors as the mean ± SD. The impact size of the Mann–Whitney U test was determined using the equation $r=Z/\sqrt{N}$, where Z is the z-score and *N* is the total size of the sample (small: 0.1; medium: 0.3; large: 0.5). Statistical significance was defined as *p* ≤ 0.05 for all analyses. Analyses were carried out by utilizing the SPSS 23 software (SPSS Inc., Chicago, IL, USA).

## 3. Results

### 3.1. Physical Activity of 6–7- and 8–9-Year-Old Children in the Experimental Group

Analysing the physical activity pre-test results of the 6–7- and 8–9-year-old children, it turned out that both the First Grade (92.15 MET, min/week) and Second Grade (97.50 MET, min/week) children in the experimental group were physically active during physical education lessons. The analysis of physical activity types, such as cycling to school, showed no differences in age, according to the MET; however, there were differences in walking to school—First Grade (15.98 MET, min/week) and Second Grade (23.50 MET, min/week)—in terms of age, according to the MET. In the context of average physical activity, a higher indicator (805.95 MET, min/week) was detected in the First Grade of the experimental group, in comparison with the Second Grade (1072.12 MET, min/week). Statistically significant differences were found in average MET for the First Grade (931.60 MET, min/week), in comparison with the Second Grade (1211.55 MET, min/week; *p* < 0.05, Table 3). The post-test of the First Grade (115.83 MET, min/week) experimental group was carried out to analyse average physical activity, in comparison with the Second Grade experimental group (130.01 MET, min/week), during physical education lessons. In the post-test, walking to school—First Grade (16.07 MET, min/week) and Second Grade (30.37 MET, min/week)—showed differences in age, according to the MET. Statistically significant differences were found during the analysis of average MET for the First Grade (1108.41 MET, min/week), in comparison with the Second Grade (1453.62 MET, min/week; *p* < 0.05, Table 3). We found a statistically significant difference between experimental and control groups (*p* < 0.05) and between pre- and post-test.

### 3.2. Physical Activity of 6–7- and 8–9-Year-Old Children in the Control Group

Analysing the results considering the physical activity of 6–7- and 8–9-year-old children, it turned out that in the control group, both the First Grade (91.68 MET, min/week) and Second Grade (95.87 MET, min/week) children were physically active in physical education lessons during the pre-test. The analysis of physical activity types, such as cycling to school, found no differences in age, according to the MET. We found that walking to school—First Grade (0.00 MET, min/week) and Second Grade (22.15 MET, min/week—showed differences in age, according to the MET. Statistically significant differences were found during the analysis of average MET for the First Grade in the control group (906.40 MET, min/week), compared to the Second Grade (1105.71 MET, min/week; *p* < 0.05, Table 4). The post-test results for the First Grade of the control group (98.10 MET, min/week) were determined by the analysis of average physical activity, in comparison with the Second Grade children of the same group (105.70 MET, min/week), when doing physical education lessons. Statistically significant differences were found in average MET for the First Grade (995.66 MET, min/week), in comparison with the Second Grade (1211.70 MET, min/week; *p* < 0.05, Table 4).

The study performed at the beginning of the experiment showed that in the pre-test, the level of somatic anxiety of the primary school children in the CG was average (4.95 ± 1.10 points). When exploring the results of the somatic anxiety in the EG (4.95 ± 1.10 points) before and after the experiment, after the intervention programme, somatic anxiety in the EG was 4.55 ± 1.00 points, indicating lower levels of depression, seclusion, somatic complaints, aggression, and delinquent behaviours (F = 4.785, *p* < 0.05, P = 0.540; Figure 1a). 

### 3.3. Anxiety of 6–7-Year-Old Children (First Grade)

When dealing with the personality anxiety results, we established that in the pre- and post-tests, the results of CG students did not statistically significantly differ (3.63 ± 0.80 points and 3.48 ± 0.50 points, respectively; F = 0.139, *p* > 0.05, P = 0.041). When analysing EG personality anxiety results in the pre- and post-tests, after the intervention programme, the EG personality anxiety results significantly decreased (3.55 ± 1.10 points and 2.78 ± 0.90 points, respectively; F = 5.195, *p* < 0.05, P = 0.549; Figure 1b). 

In the pre-test, the level of social anxiety in the CG was 6.15 ± 1.30 points. The post-test CG result was statistically significantly lower (5.18 ± 1.20 points; F = 4.75, *p* < 0.05, P = 0.752). When analysing the levels of the social anxiety of the EG, pre- and post-test results decreased after the intervention programme (6.32 ± 1.10 points and 4.25 ± 1.40 points, respectively) and significantly differed (F = 8.029, *p* < 0.05, P = 0.673; Figure 1c).

### 3.4. Anxiety of 8–9-Year-Old Children (Second Grade) 

The research performed at the beginning of the experiment showed that in the pre-test, the level of somatic anxiety of the adolescents in the CG was average (4.63 ± 1.10 points). When exploring the somatic anxiety results in the EG (4.50 ± 0.90 points) before the experiment and after it, a decrease in somatic anxiety in the EG was established (4.10 ± 0.75 points), indicating lower levels of depression, seclusion, somatic complaints, aggression, and delinquent behaviours (F = 4.482, *p* < 0.05, P = 0.610; Figure 1a). 

When dealing with the personality anxiety results, we established that in the pre- and post-test, the results of CG students were not statistically significantly different (3.10 ± 0.85 points and 2.86 ± 0.67 points, respectively; F = 0.127, *p* > 0.05, P = 0.057). When analysing the pre- and post-test EG personality anxiety results, after the intervention programme, the EG personality anxiety results decreased (2.93 ± 0.93 points vs. 2.51 ± 1.00 points, respectively; F = 6.498, *p* < 0.05, P = 0.758; Figure 1b). 

In the pre-test, the level of social anxiety in the CG was 4.55 ± 1.30 points. The post-test CG result was statistically significantly lower (3.70 ± 1.40 points; F = 4.218, *p* < 0.05, P = 0.652). When analysing the levels of social anxiety in the EG, pre- and post-test results decreased after the intervention programme (4.65 ± 1.15 points and 3.01 ± 1.50 points, respectively) and were significantly different (F = 8.021, *p* < 0.05, P = 0.798; Figure 1c). 

## 4. Discussion

The outcomes of this study showed that the proposed procedure for a physical education program and educational model encouraging physical activity in children had an impact on three primary dimensions—somatic anxiety, personality anxiety, and social anxiety—for children aged 6–7 and 8–9 years. The procedure depended on dynamic exercise, intense motor skills reiteration, differentiation, seating and parking reduction, and physical activity dissemination in the classroom model. Following eight months of applying this study’s physical education program, anxiety decreased in the children. Schools provide an opportune site for addressing PA promotion in children. With children spending a substantial number of their waking hours during the week at school, increased opportunities for PA are needed, especially considering trends toward decreased frequency of physical education in schools [31,32]. Considering physical education curricula, Chen et al. [29] described the following: Aerobic activities: Most daily activities should be moderate- to vigorous-intensity aerobic activities, such as bicycling, playing sports and active games, and brisk walking.Strength training: The program should include muscle-strengthening activities at least three days a week, such as performing calisthenics, weight-bearing activities, and weight training.Bone strengthening: Bone-strengthening activities should also be included at least three days a week, such as jump-rope, playing tennis or badminton, and engaging in other hopping-type activities.

School-related physical activity interventions may reduce anxiety, increase resilience, improve well-being, and increase positive mental health in children and adolescents [33]. Increasing activity levels and sports participation among the least active young people should be a target of community- and school-based interventions in order to promote well-being. Frequency of physical activity has been positively correlated with well-being and negatively correlated with both anxiety and depressive symptoms, up to a threshold of moderate frequency of activity. In a multi-level mixed effects model, more frequent physical activity and participation in sport were both found to independently contribute to greater well-being and lower levels of anxiety and depressive symptoms in both sexes [34]. There does not appear to be an additional benefit to mental health associated with meeting the WHO-recommended levels of activity [9]. Physical activity interventions have been shown to have a small beneficial effect in reducing anxiety; however, the evidence base is limited. Reviews of physical activity and cognitive functioning have provided evidence that routine physical activity can be associated with improved cognitive performance and academic achievement, but these associations are usually small and inconsistent [35]. Advances in neuroscience have resulted in substantial progress in linking physical activity to cognitive performance, as well as to brain structure and function [36]. The executive functions hypothesis proposes that exercise has the potential to induce vascularization and neural growth and alter synaptic transmission in ways that alter thinking, decision making, and behaviour in those regions of the brain tied to executive functions—in particular, the pre-frontal cortices [37,38]. The brain may be particularly sensitive to the effects of physical activity during pre-adolescence, as the neural circuitry of the brain is still developing [8]. 

During their school years, about 33% of primary and secondary school students experience the adverse effects of test anxiety [39]. Anxiety is an aversive motivational state which occurs when the degree of perceived threat is viewed as high [40]. In the concept of anxiety, a frequently made differentiation is created between trait anxiety, referring to differences in personality dimensions, and state anxiety, alluding to anxiety as a transient mindset state. These two kinds of anxiety hamper performance, particularly during complex and intentionally requested assignments [41]. Mavilidi et al. [42] presented a study investigating whether a short episode of physical activity can mitigate test anxiety and improve test execution in 6th grade children (11–12 years). The discoveries of the study by the above authors expressed that, even though test anxiety was not decreased as expected, short physical activity breaks can be utilized before assessments without blocking academic performance [43].

Physical activity has been associated with physiological, developmental, mental, cognitive, and social health benefits in young people [36]. While the health benefits of physical activity are well-established, higher levels of physical activity have also been associated with enhanced academic-related outcomes, including cognitive function, classroom behaviour, and academic achievement [44]. The evidence suggests a decline in physical activity from early childhood [45]. The physical and psychological benefits of physical activity for children and adolescents include reduced adiposity and cardiometabolic risk factors, as well as improvements in musculoskeletal health and psychological well-being [33,46,47]. However, population based-studies have reported that more than half of all children internationally are not meeting the recommended levels of physical activity, with rates of compliance declining with age from the early primary school years [9]. Therefore, it is imperative to promote physical activity and intervene early in childhood, prior to such a decline in physical activity [48]. Schools are considered ideal settings for the promotion of children’s physical activity. There are multiple opportunities for children to be physically active over the course of the school week, including during break times, sport, physical education class, and active travel to and from school [49]. There exists strong evidence of the benefits of physical activity for the mental health of children and adolescents, mainly in terms of depression, anxiety, self-esteem, and cognitive functioning [35]. 

Physiological adaptation (e.g., hormonal regulation) of the body during physical exercise can be applied additionally to psychosocial stressors, thus improving mental health [48]. Subsequently, it has been stated that intense physical activity which improves health-related fitness may be expected to evoke neurobiological changes affecting psychological and academic performance [43].

The results of this review contribute to knowledge about the multifaceted interactions influencing how physical activity can be enhanced within a school setting, given certain contexts. Evidence has indicated that school-based interventions can be effective in enhancing physical activity, cardiorespiratory and muscular fitness, psychosocial outcomes associated with physical activity (e.g., enjoyment), and other markers of health status in children. School- and community-based physical activity interventions, as part of an obesity prevention or treatment programme, can benefit the executive functions of children, specifically those with obesity or who are overweight [46]. Considering the positive effects of physical activity on health in general, these findings may reinforce school-based initiatives to increase physical activity [34]. This involves classroom teachers incorporating physical activity into class time, either by integrating physical activity into physically active lessons, or adding short bursts of physical activity with curriculum-focused active breaks [50,51]. It is widely accepted that physical inactivity is an important risk factor for chronic diseases; prevention strategies should begin as early as childhood, as the prevalence of physical inactivity increases even more in adolescence [52]. A physically active lifestyle begins to form very early in childhood and has a positive tendency to persist throughout life [52].

We all have an important role to play in increasing children’s physical activity. Schools must promote and influence a healthy environment for children. Most primary school children spend an average of 6–7 h a day at school, which is most of their daytime. A balanced and adapted physical education lesson provides cognitive content and training for developing motor skills and knowledge in the field of physical activity. Our 8-month physical education program can give children the opportunity to increase physical activity and improve emotional well-being, which can encourage children to be physically active throughout life.

## 5. Conclusions

Low physical activity in children is a major societal problem. The growing number of children with obesity is a concern for doctors and scientists. The focus of our study was to improve emotional well-being and physical activity in children. Since elementary school children spend most of their day at school, physical education lessons are a great tool to increase physical activity. A balanced and adapted physical education lesson can help to draw children’s attention to the health benefits of physical activity. It was established that the properly constructed and purposefully applied 8-month physical education program had an impact on the physical activity and emotional well-being of primary school children (i.e., 6–7 and 8–9 year olds) in three main dimensions: somatic anxiety, personality anxiety, and social anxiety. Our findings suggest that the 8-month physical education program intervention is effective for increasing levels of physical activity. Changes in these activities may require more intensive behavioural interventions in children or upstream interventions at the family and societal level, as well as at the school environment level. These findings have relevance for researchers, policy makers, public health practitioners, and doctors who are involved in health promotion, policy making, and commissioning services.

## Figures and Tables

**Figure 1 ijerph-18-07536-f001:**
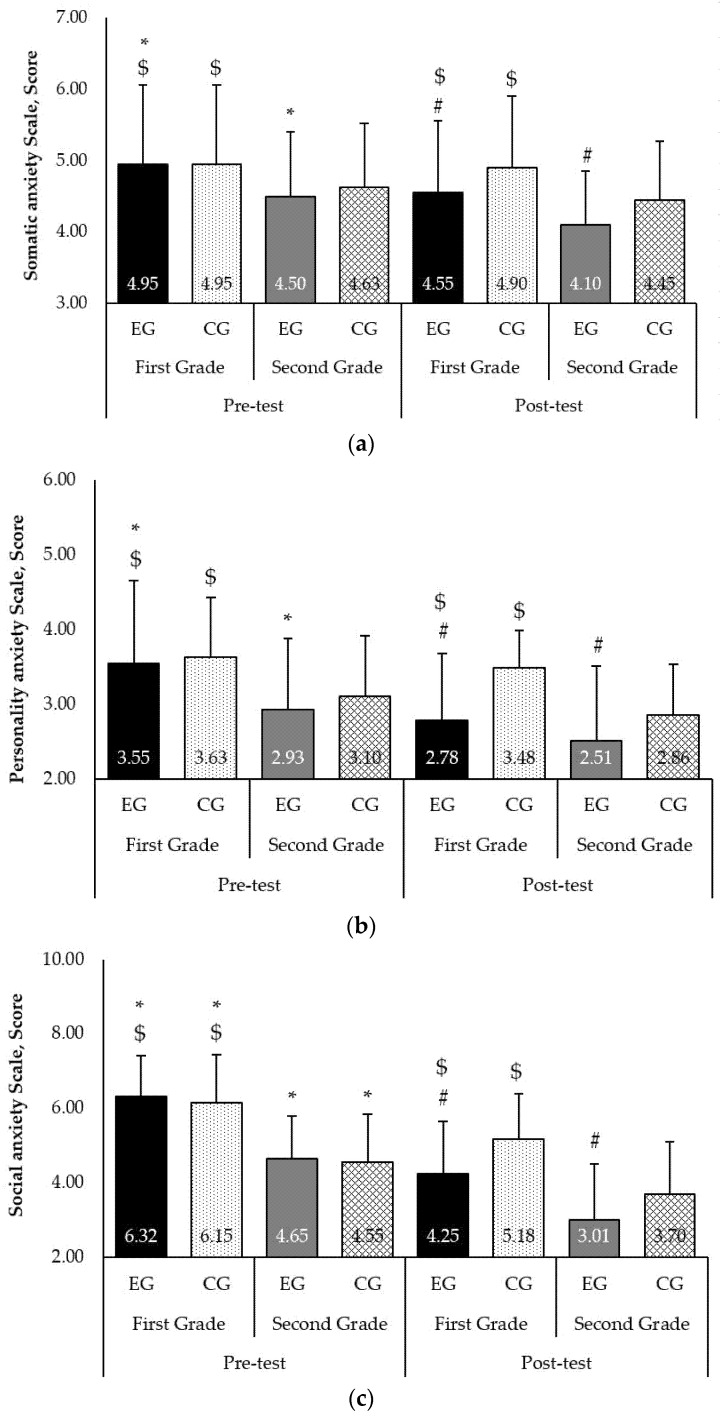
Pre- and post-test levels of somatic anxiety (**a**), personality anxiety (**b**), and social anxiety (**c**) in primary school children. ^#^, *p* < 0.05 between experimental and control groups; ^$^, *p* < 0.05 between First and Second Grades; *, *p* < 0.05 between pre- and post-test.

**Table 1 ijerph-18-07536-t001:** Dynamic exercise, intense motor skills reiteration, differentiation, seating and parking reduction, and physical activity distribution in the classroom (DIDSFA) model—expanding dynamic learning time in primary physical education.

Dynamic exercise	Aerobic capacity and/or muscle strength education. Exercise can be any movement that improves physical fitness. Exercise that gives you more energy or stamina is regularly called aerobic exercise [26].
Intense motor skills repetition	Diminishing/eliminating queues, such that children are not waiting their turn; having small-sided games or group activities like 3 versus 3 (which expands the number of times the children need to develop/apply their skills—this assists in preventing children from being on the periphery of or excluded from a game/activity); and expanding the amount of gear equipment for the children and/or potentially expanding the number of stations.
Differentiation	All children ought to be set assignments that are fit to their physical, intellectual, and social situation, which encourages them to take part in active learning time. Teachers ought to guarantee that they know about the space, errand, hardware, and individuals (STEP) structure for the dynamic differentiation of activities [28,29].
Seating and parking reduction	When a teacher is providing feedback or questioning students frequently, they do not have to stop the entire class; instead, they can simply target and stop a group of students or an individual child. Engaging children in activity quickly toward the beginning of the exercise, through concise questioning and feedback. Guaranteeing that equipment is prepared, coordinated, and available toward the beginning of and all through the exercise [27].
Physical activity distribution in the classroom	This rule depends on teachers encouraging children’s in-class physical activity through positive praise. Instances of the advancement of in-class physical activity incorporate “incredible collaboration, continue moving, and searching for space” [28].

**Table 2 ijerph-18-07536-t002:** Physical education program (First and Second Grades).

	Lesson Topic	Areas of Activity for the Physical Education Lesson	Lesson Topic	Areas of Activity for the Physical Education Lesson
	First Grade		Second Grade	
Month 1	Exercising with a textbook and notes. Arrangement, basic starting hand and leg positions. Honest conduct. Proper breathing over time.	Healthy lifestyle Movement skills Healthy lifestyle Sport units (athletics)	Working with a textbook and notes. Walking and running exercises. Smooth running in a group. 3 × 10 m speed shuttle run test—agility. Playing with balls.	Healthy lifesty leMovement skills Sport units (athletics) Sport units (sports games)
Month 2	Exercising with a textbook and notes. Ball school. I pass the ball to a companion. I am figuring out how to pass the ball precisely. Running is the best movement. Running: Relay. Proper posture.	Healthy lifestyle Sport units (basketball) Sport units (athletics)	Working with a textbook and notes. Exercises with ball. Football game. Flexibility training. Developing movement skills through play.	Healthy lifestyle Movement skills Sport units (football) Sport units (gymnastics)
Month 3	Exercising with a textbook and notes. Jumping on two feet. Spider and turn. Animal aerobatics. Let us jump by jumping. Shuttle running 3 × 10 m.	Healthy lifestyle Movement skills Unconventional physical activity Sport units (athletics)	Working with a textbook and notes. Let us get acquainted with game of square. Let us learn to play square.Long jump-rope. Activity games. The long jump test to test explosive power of children’s leg muscles.	Healthy lifestyle Movement skills Sport units (gymnastics) Sport units (athletics)
Month 4	Exercising with a textbook and notes. Long jump. Figure out how to kick and drive a soccer ball, to drive a soccer ball in a straight and winding line. Children’s yoga.	Healthy lifestyle Sport units (athletics) Sport units (football) Unconventional physical activity	Working with a textbook and notes. Exercises to help calm down and concentrate. Kids yoga. Throw a small ball at a target (vertical). Throw a small ball at a target (horizontal). Throw a small ball at a target (vertical and horizontal).	Healthy lifestyle Unconventional physical activity Movement skills Sport units (athletics)
Month 5	Exercising with a textbook and notes. How to kick a soccer ball into the goal. The basics of aerobatics: practice with gymnastic balls. Muscle stretching. Attempt to keep balance.	Healthy lifestyle Sport units (football) Unconventional physical activity	Working with a textbook and notes. Overcoming horizontal and vertical barriers. Jumping. Hanging. Medical (stuffed) 1 kg ball pushing from the chest to test the explosive power of hands. Aerobics. Movement skills outdoors.	Healthy lifestyle Movement skills Sport units (athletics) Unconventional physical activity
Month 6	Exercising with a textbook and notes. Basics of gymnastics implies tools and right posture. Jump-rope. Basic strides of aerobics. Fun relays.	Healthy lifestyle Sport units (gymnastics) Unconventional physical activity Movement skills	Working with a textbook and notes. Getting ready and learning to play basketball. Learn to rotate gymnastics hoop. Methods of movements in space (darkness).	Healthy lifestyle Sport units (basketball) Sport units (gymnastics) Unconventional physical activity
Month 7	Exercising with a textbook and notes. We figure out how to drive, pass, and catch a basketball by exercising in pairs, to drive a basketball in a straight and winding line. Obstacle course.	Healthy lifestyle Sport units (basketball) Movement skills Unconventional physical activity	Working with a textbook and notes. Playing with balls. How to move a log without falling. Running from a high start. Running from a low start. Starting positions (high or low start).	Healthy lifestyle Movement skills Sport units (athletics) Unconventional physical activity
Month 8	Exercising with a textbook and notes. Tossing a ball. We cooperate to overcome obstacles. We play football. We figure out how to orient ourselves. Sports event.	Healthy lifestyle Sport units (athletics) Sport units (football) Movement skills	Working with a textbook and notes. Outdoor games. We learn how to orient in the area. Strengthening the musculoskeletal system. Outdoor Project “Health and Sport Day”.	Healthy lifestyle Movement skills Sport units (gymnastics) Unconventional physical activity

**Table 3 ijerph-18-07536-t003:** Physical activity levels determined using the MET method.

Type of Physical Activity	Physical Education Lesson	Cycling to School	Walking to School	Sport Groups (Mean Physical Activity)
MET	3.5	4	3.3	6
1 day/min	30	0.45	0.3	59
Days per week	1	3	4	1
	**MET, min/week**			
The experimental group	Pre-test		Post-test	
Grade	First	Second	First	Second
Physical Education lesson	92.15	97.50	115.83	130.01
Cycling to school	17.52	18.40	18.39	21.33
Walking to school	15.98	23.50	16.07	30.37
Sport groups (mean physical activity)	805.95	1072.12	958.12	1271.91
On average	931.60 * ^#^ ^$ §^	1211.55 * ^#^ ^§^	1108.41 * ^#^ ^$^	1453.62 * ^#^

Note. *, *p* < 0.05 (according to the Mann–Whitney U test) between physical activity types; ^#^, *p* < 0.05 (according to the Mann–Whitney U test) between experimental and control groups; ^$^, *p* < 0.05 (according to the Mann–Whitney U test) between First and Second Grades; ^§^, *p* < 0.05 (according to the Mann–Whitney U test) between pre-test and post-test.

**Table 4 ijerph-18-07536-t004:** The physical activity level using the MET method (the pre-test/post-test results of the control group).

Type of Physical Activity	Physical Education Lesson	Cycling to School	Walking to School	Sport Groups (Mean Physical Activity)
MET	3.5	4	0	6
1 day/min	30	0.58 *|0.50 **	0.3 *|0.71 **	58
Days per week	1	3	4	1
Note. *—First Grade; **—Second Grade.				
	**MET, min/week**			
The control group	Pre-test		Post-test	
Grade	First	Second	First	Second
Physical Education lesson	91.68	95.87	98.1	105.7
Cycling to school	15.91	23.03	16.58	23.54
Walking to school	0	22.15	0	28.65
Sport groups (mean physical activity)	798.81	964.66	880.98	1053.81
On average	906.40 * ^$^	1105.71 *	995.66 * ^$^	1211.70 *
Note. *, *p* < 0.05 (according to the Mann–Whitney U test) between physical activity types; ^$^, *p* < 0.05 (according to the Mann–Whitney U test) between First and Second Grades.				

## Data Availability

The data presented in this study are available on request from the corresponding author.

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
