# Peer review of "Effects of a Physical Education Program on Physical Activity and Emotional Well-Being among Primary School Children"

_ijerph, 2021, doi:10.3390/ijerph18147536_

Round 1

Reviewer 1 Report

Review for Manuscript entitled “Effects of a Physical Education Program on Physical Activity and Emotional Well-being Among primary school children”.

Thank you for providing me with the opportunity to review the manuscript entitled “Effects of a Physical Education Program on Physical Activity and Emotional Well-being Among primary school children”.

Introduction:

The introduction provides an overview. In addition, introduction is written clearly and is interesting to read.

Materials and Methods:

Do the study is registrationed in ClinicalTrials.gov? What number trial registration has?

The evaluation of physical activity: Provide the test score and psychometric properties. This will help interpret whether statistical analyses are suitable.

Data Analysis

State how homogeneity of sample was measured (e.g., histograms, Shapiro-Wilk test).

Conclusion

Consider providing section(s) on implications for future research or limitations (e.g. inclusion criteria bias).

Author Response

Reviewer 1

Review for Manuscript entitled “Effects of a Physical Education Program on Physical Activity and Emotional Well-being Among primary school children”.

 Firstly, the authors want to thank your contribution to our paper. We really appreciate you taking the time out to share your experience with us.

Thank you for providing me with the opportunity to review the manuscript entitled “Effects of a Physical Education Program on Physical Activity and Emotional Well-being Among primary school children”.

Introduction:

The introduction provides an overview. In addition, introduction is written clearly and is interesting to read.

Materials and Methods:

Do the study is registrationed in ClinicalTrials.gov? What number trial registration has?

We haven/t ClinicalTrials.gov. It is methodology of social science. This study was approved by the research ethics committee of the Kaunas University of Technology, Institute of Social Science and Humanity (Protocol No V19-1253-03).

The evaluation of physical activity: Provide the test score and psychometric properties. This will help interpret whether statistical analyses are suitable.

Data Analysis

State how homogeneity of sample was measured (e.g., histograms, Shapiro-Wilk test).

Table 3. Physical activity levels determined using the MET method (pre-/post-test *,

Type of physical activity

Physical Education lesson

Cycling to school

Walking to school

Sport groups
(mean physical activity)

MET

3.5

4

3.3

6

1 day/min

30

0.45

0.3

59

Days per week

1

3

4

1

MET, min/week

The experimental group

Pre-test

Post-test

Grade

First

Second

First

Second

Physical Education lesson

92.15

97.50

115.83

130.01

Cycling to school

17.52

18.40

18.39

21.33

Walking to school

15.98

23.50

16.07

30.37

Sport groups (mean physical activity)

805.95

1072.12

958.12

1271.91

On average

931.60* # $ §

1211.55* # §

1108.41* # $

1453.62* #

*, p < 0.05 (according to the Mann–Whitney U-test) between physical activity types; #, p < 0.05 (according to the Mann–Whitney U-test) between Experimental and Control groups; $, p < 0.05 (according to the Mann–Whitney U-test) between First and Second Grades; §, p < 0.05 (according to the Mann–Whitney U-test) between Pre-test and Post-test.

Table 4. The physical activity level using the MET method (the pre-test/post-test results of the control group)

Type of physical activity

Physical Education lesson

Cycling to school

Walking to school

Sport groups
(mean physical activity)

MET

3.5

4

0

6

1 day/min

30

0.58* | 0.50**

0.3* | 0.71**

58

Days per week

1

3

4

1

* – First Grade

** – Second Grade

MET, min/week

The control group

Pre-test

Post-test

Grade

First

Second

First

Second

Physical Education lesson

91.68

95.87

98.10

105.70

Cycling to school

15.91

23.03

16.58

23.54

Walking to school

0.00

22.15

0.00

28.65

Sport groups (mean physical activity)

798.81

964.66

880.98

1053.81

On average

906.40* $

1105.71*

995.66* $

1211.70*

Note. *, p < 0.05 (according to the Mann–Whitney U-test) between physical activity types; $, p < 0.05 (according to the Mann–Whitney U-test) between First and Second Grades.

Conclusion

Consider providing section(s) on implications for future research or limitations (e.g. inclusion criteria bias).

Although schools can offer unique opportunities for structured physical activity for children, there is a tendency to cut back physical education lessons. Lithuanian scientists propose new methodology of PE. Physical education activities were taught through physical schooling, by preparing a textbook comprising two inter-related parts: a textbook, children’s notes, teachers’ book. We want perform longitudinal studies.

  1. Klizas, Š.; Klizienė, I.; Gruodytė-Račienė, R.; Buliuolienė, L.; Daukšaitė, L. Physical Education: textbook First Grade. Kaunas : Šviesa, 2016.

Buliuolienė, L.; Daukšaitė, L.; Klizas, Š.; Klizienė, I.; Cibulskas, G. Physical Education: textbook Second Grade. Kaunas : Šviesa, 2017.

Reviewer 2 Report

Introduction

The introduction is well written and clearly. The sense of discourse is also clear, but I find two major problems.

The first concerns the confusion between physical education and physical activity. At the beginning of the introduction, you talk about the methodological and didactic changes that Physical Education has had in the last 50 years. But the subject of her research is not Physical Education, but rather an intervention program to increase physical activity in elementary school children. In addition, introducing a physical activity program in the Physical Education subject is not a methodological change, I even consider that it is a limitation since you exclude other contents that are also typical of the subject such as body expression and aesthetics, activities in nature or games to focus your attention on sports activities and health.

Later you indicate that Physical Education (I understand that you are referring to the subject) can cause negative feelings and anxiety because it has a competitive, comparative and "natural" selection perspective. They point out about the Physical Education subject is true. However, you de-ideologize and depoliticize physical activity, or rather, the choice of content in your intervention program to increase physical activity. You select football, basketball, gymnastics and athletics as fundamental activities of your intervention. However, these activities are competitive, comparative and, in a way, require certain biological characteristics to successfully play these sports.

Why do they show the negative aspects of Physical Education and not the negative aspects of the choices they make for their physical activity program?

The second problem I find is that the subject is a large and deep expliand in previous bibliography on related the benefits of physical activity programs in children and adolescents. The contributions that you make in article are limited since the beneficial effects of this type of program have been previously described. I have not been able to find something new in your work that makes it different from what has already been published previously.

Materials and methods

This point is one of the best aspects of the article. The authors explain in detail the elements and procedures of their research. I have only found two small aspects to improve. In lines 185 to 189 the same phrase is repeated:

The physical education pack considers a "natural" 185

kind of integration and dynamic learning, building awareness, encouraging sensitivity to 186

nature, and supporting healthy styles of living. The physical education pack takes into 187

consideration a "natural" kind of integration and dynamic learning, building awareness, 188

encouraging sensitivity to nature, and supporting healthy styles of living 199

I think it is important to indicate the academic year in which they have carried out this intervention.

Results

The tables with the results are not very clear and I had to put a lot of effort to understand how it was structured. Perhaps restructuring the tables can make it easier for the reader to understand.

Discussion and conclusions

In this sense, to what was commented in the introduction, the article does not make significant contributions and the discussion confirms what has already been indicated by similar studies.

Author Response

Reviewer 2

Firstly, the authors want to thank your contribution to our paper. We really appreciate you taking the time out to share your experience with us.

The introduction is well written and clearly. The sense of discourse is also clear, but I find two major problems.

The first concerns the confusion between physical education and physical activity. At the beginning of the introduction, you talk about the methodological and didactic changes that Physical Education has had in the last 50 years. But the subject of her research is not Physical Education, but rather an intervention program to increase physical activity in elementary school children. In addition, introducing a physical activity program in the Physical Education subject is not a methodological change, I even consider that it is a limitation since you exclude other contents that are also typical of the subject such as body expression and aesthetics, activities in nature or games to focus your attention on sports activities and health.

Later you indicate that Physical Education (I understand that you are referring to the subject) can cause negative feelings and anxiety because it has a competitive, comparative and "natural" selection perspective. They point out about the Physical Education subject is true. However, you de-ideologize and depoliticize physical activity, or rather, the choice of content in your intervention program to increase physical activity. You select football, basketball, gymnastics and athletics as fundamental activities of your intervention. However, these activities are competitive, comparative and, in a way, require certain biological characteristics to successfully play these sports.

Why do they show the negative aspects of Physical Education and not the negative aspects of the choices they make for their physical activity program?

Response:

Physical education has been a part of school curriculums for many years, but, due to childhood obesity, focus has increased on the role that schools play in physical activity and monitoring physical fitness (Ogden, C.L.; Carroll, M.D.; Curtin, L.R.; McDowell, M.A.; Tabak, C.J.; Flegal, K.M. Prevalence of overweight and obesity in the United States, 19992004. JAMA 2006, 295, 15491555; Kelder, S.H.; Springer, A.E.; Barroso, C.S.; Smith, C.L.; Sanchez, E.; Ranjit, N.; Hoelscher, D.M. (2009). Implementation of Texas Senate Bill 19 to increase physical activity in elementary schools. J. Public Health Policy 2009, 30, 221247). Although schools are able to offer unique opportunities for structured physical activity for children, there is a tendency to cut back physical education lessons. After the research “Effects of a Physical Education Program on Physical Activity and Emotional Well-being Among primary school children” we prepared new guidelines for innovative physical education lessons and new PE methodology. New methodology of PE is described: physical education activities were taught through physical schooling, by preparing a textbook comprising two inter-related parts: a) a textbook and b) children’s notes. The textbooks were filled with logical tasks, self-evaluation, and activities relating to spatial perception and self-improvement. The methodological devices provide strategies for practicing with textbooks. The physical education pack considers a "natural" kind of integration and dynamic learning, building awareness, encouraging sensitivity to nature, and supporting healthy styles of living. The physical education pack takes into consideration a "natural" kind of integration and dynamic learning, building awareness, encouraging sensitivity to nature, and supporting healthy style of living. The instructor's manual has a unified structure, which makes it simple to utilize. Its proposals and advice are clear. The advanced version helps educators in their planning and execution activities.

Fig. 1. Methodical material for physical education lessons

  1. Ogden, C.L.; Carroll, M.D.; Curtin, L.R.; McDowell, M.A.; Tabak, C.J.; Flegal, K.M. Prevalence of overweight and obesity in the United States, 1999– JAMA 2006, 295, 1549–1555.
  2. Kelder, S.H.; Springer, A.E.; Barroso, C.S.; Smith, C.L.; Sanchez, E.; Ranjit, N.; Hoelscher, D.M. (2009). Implementation of Texas Senate Bill 19 to increase physical activity in elementary schools. Public Health Policy 2009, 30, 221–247.

The second problem I find is that the subject is a large and deep explained in previous bibliography on related the benefits of physical activity programs in children and adolescents. The contributions that you make in article are limited since the beneficial effects of this type of program have been previously described. I have not been able to find something new in your work that makes it different from what has already been published previously.

 Response:

Novelty of work: For the first time, PE curriculum has been developed for second grade children’s, a new approach to physical education methodology. For the first time anxiety is measured between first and second grades.

Materials and methods

This point is one of the best aspects of the article. The authors explain in detail the elements and procedures of their research. I have only found two small aspects to improve. In lines 185 to 189 the same phrase is repeated:

The physical education pack considers a "natural" 185

kind of integration and dynamic learning, building awareness, encouraging sensitivity to 186

nature, and supporting healthy styles of living. The physical education pack takes into 187

consideration a "natural" kind of integration and dynamic learning, building awareness, 188

encouraging sensitivity to nature, and supporting healthy styles of living 199

I think it is important to indicate the academic year in which they have carried out this intervention.

The intervention was caried out from September 2018 to May 2019

Results

The tables with the results are not very clear and I had to put a lot of effort to understand how it was structured. Perhaps restructuring the tables can make it easier for the reader to understand.

Response:

It was corrected

 Table 3. Physical activity levels determined using the MET method (pre-/post-test *, p < 0.05 (according to the Mann–Whitney U-test) between physical activity types; #, p < 0.05 (according to the Mann–Whitney U-test) between Experimental and Control groups; $, p < 0.05 (according to the Mann–Whitney U-test) between First and Second Grades; §, p < 0.05 (according to the Mann–Whitney U-test) between Pre-test and Post-test.

Type of physical activity

Physical Education lesson

Cycling to school

Walking to school

Sport groups
(mean physical activity)

MET

3.5

4

3.3

6

1 day/min

30

0.45

0.3

59

Days per week

1

3

4

1

MET, min/week

The experimental group

Pre-test

Post-test

Grade

First

Second

First

Second

Physical Education lesson

92.15

97.50

115.83

130.01

Cycling to school

17.52

18.40

18.39

21.33

Walking to school

15.98

23.50

16.07

30.37

Sport groups (mean physical activity)

805.95

1072.12

958.12

1271.91

On average

931.60* # $ §

1211.55* # §

1108.41* # $

1453.62* #

Table 4. The physical activity level using the MET method (the pre-test/post-test results of the control group)

Type of physical activity

Physical Education lesson

Cycling to school

Walking to school

Sport groups
(mean physical activity)

MET

3.5

4

0

6

1 day/min

30

0.58* | 0.50**

0.3* | 0.71**

58

Days per week

1

3

4

1

* – First Grade

** – Second Grade

MET, min/week

The control group

Pre-test

Post-test

Grade

First

Second

First

Second

Physical Education lesson

91.68

95.87

98.10

105.70

Cycling to school

15.91

23.03

16.58

23.54

Walking to school

0.00

22.15

0.00

28.65

Sport groups (mean physical activity)

798.81

964.66

880.98

1053.81

On average

906.40* $

1105.71*

995.66* $

1211.70*

Note. *– p<0.05 (according to the Mann-Whitney U test) between physical activity types; $– p<0.05 (according to the Mann-Whitney U test) between First and Second Grades.

Discussion and conclusions

In this sense, to what was commented in the introduction, the article does not make significant contributions and the discussion confirms what has already been indicated by similar studies.

Although schools can offer unique opportunities for structured physical activity for children, there is a tendency to cut back physical education lessons. Lithuanian scientists propose new methodology of PE. Physical education activities were taught through physical schooling, by preparing a textbook comprising two inter-related parts: a textbook, children’s notes, teachers’ book.

  1. Klizas, Š.; Klizienė, I.; Gruodytė-Račienė, R.; Buliuolienė, L.; Daukšaitė, L. Physical Education: textbook First Grade. Kaunas : Šviesa, 2016.
  2. Buliuolienė, L.; Daukšaitė, L.; Klizas, Š.; Klizienė, I.; Cibulskas, G. Physical Education: textbook Second Grade. Kaunas : Šviesa, 2017.

Round 2

Reviewer 2 Report

Introduction

Original (review 1)

The introduction is well written and clearly. The sense of discourse is also clear, but I find two major problems.

The first concerns the confusion between physical education and physical activity. At the beginning of the introduction, you talk about the methodological and didactic changes that Physical Education has had in the last 50 years. But the subject of her research is not Physical Education, but rather an intervention program to increase physical activity in elementary school children. In addition, introducing a physical activity program in the Physical Education subject is not a methodological change, I even consider that it is a limitation since you exclude other contents that are also typical of the subject such as body expression and aesthetics, activities in nature or games to focus your attention on sports activities and health.

Later you indicate that Physical Education (I understand that you are referring to the subject) can cause negative feelings and anxiety because it has a competitive, comparative and "natural" selection perspective. They point out about the Physical Education subject is true. However, you de-ideologize and depoliticize physical activity, or rather, the choice of content in your intervention program to increase physical activity. You select football, basketball, gymnastics and athletics as fundamental activities of your intervention. However, these activities are competitive, comparative and, in a way, require certain biological characteristics to successfully play these sports.

Why do they show the negative aspects of Physical Education and not the negative aspects of the choices they make for their physical activity program?

 Response authors:

Physical education has been a part of school curriculums for many years, but, due to childhood obesity, focus has increased on the role that schools play in physical activity and monitoring physical fitness (Ogden, C.L.; Carroll, M.D.; Curtin, L.R.; McDowell, M.A.; Tabak, C.J.; Flegal, K.M. Prevalence of overweight and obesity in the United States, 1999–2004. JAMA 2006, 295, 1549–1555; Kelder, S.H.; Springer, A.E.; Barroso, C.S.; Smith, C.L.; Sanchez, E.; Ranjit, N.; Hoelscher, D.M. (2009). Implementation of Texas Senate Bill 19 to increase physical activity in elementary schools. J. Public Health Policy 2009, 30, 221–247).

Although schools are able to offer unique opportunities for structured physical activity for children, there is a tendency to cut back physical education lessons. After the research “Effects of a Physical Education Program on Physical Activity and Emotional Well-being Among primary school children” we prepared new guidelines for innovative physical education lessons and new PE methodology. New methodology of PE is described: physical education activities were taught through physical schooling, by preparing a textbook comprising two inter-related parts: a) a textbook and b) children’s notes. The textbooks were filled with logical tasks, self-evaluation, and activities relating to spatial perception and self-improvement. The methodological devices provide strategies for practicing with textbooks. The physical education pack considers a "natural" kind of integration and dynamic learning, building awareness, encouraging sensitivity to nature, and supporting healthy styles of living. The physical education pack takes into consideration a "natural" kind of integration and dynamic learning, building awareness, encouraging sensitivity to nature, and supporting healthy style of living. The instructor's manual has a unified structure, which makes it simple to utilize. Its proposals and advice are clear. The advanced version helps educators in their planning and execution activities.

Fig. 1. Methodical material for physical education lessons

  1. Ogden, C.L.; Carroll, M.D.; Curtin, L.R.; McDowell, M.A.; Tabak, C.J.; Flegal, K.M. Prevalence of overweight and obesity in the United States, 1999–2004. JAMA 2006, 295, 1549–1555.
  2. Kelder, S.H.; Springer, A.E.; Barroso, C.S.; Smith, C.L.; Sanchez, E.; Ranjit, N.; Hoelscher, D.M. (2009). Implementation of Texas Senate Bill 19 to increase physical activity in elementary schools. J. Public Health Policy 2009, 30, 221–247.

Response to reviewer (review 2)

Childhood obesity is a global problem and is it a pandemic problem, we agree. However, the fact that physical education (PE) is the only structured activity that children have access to for physical activity is not a strong argument to justify this type of program in the school setting. This refers to a structural problem of a social nature since the system is not capable of generating time-spaces for the practice of physical activity for children and adolescents. But it is not an educational problem. The program that you present at least from the educational point of view since there is no intrinsic didactic objective to the subject itself.  

In this sense, you address a social problem where the objective is also social. You could talk about linking the school with the social context, which would give more power to your work, but it is not appropriate to link it with PE subject. Therefore, in the previous review I indicated that you confuse the concepts of the intervention program with the PE subject. Confusion that they continue to make.

On the other hand, you point out that "we prepared new guidelines for innovative physical education lessons and new PE methodology", but these innovative guidelines are materialized in a textbook. The textbook is not an innovation, in fact, it is the most widely used and widespread teaching material among teachers. This is a didactic material that encourages forms of learning based on repetition and uniformity. In addition, they indicate that there is an advanced version of the textbook. Does that mean that there is a non-advanced version? What is the difference between the two versions? Do you commercially sell an advanced version?

Finally, and regarding this point, in the first review I asked you a question about why you indicated that physical education lessons had a negative perception due to competition, comparison and "natural" selection that you indicated. I was concerned about the de-ideologization and depoliticization of physical activity even though the activities were football, basketball, gymnastics and athletics, which are fundamentally competitive, comparative and require innate biological aspects to be more successful. I need you to answer this question.

Original (review 1)

The second problem I find is that the subject is a large and deep explained in previous bibliography on related the benefits of physical activity programs in children and adolescents. The contributions that you make in article are limited since the beneficial effects of this type of program have been previously described. I have not been able to find something new in your work that makes it different from what has already been published previously.

Response:

Novelty of work: For the first time, PE curriculum has been developed for second grade children’s, a new approach to physical education methodology. For the first time anxiety is measured between first and second grades.

Response to reviewer (review 2)

I do not understand this answer because of the didactic implications it has. If the novelty is that the PE curriculum is developed, what did the children of second grade do before applying this program? Nothing? Wasn't there a curriculum that the teachers developed? I am struck by this regarding the first sentence of your response to my first review as you claim that the PE curriculum has been developing for some years.

Materials and methods; and results are clearly. Thanks for changing the results shown in the tables. Now they are clearer.

Discussion and conclusions

Original (review 1)

In this sense, to what was commented in the introduction, the article does not make significant contributions and the discussion confirms what has already been indicated by similar studies.

Reponse to authors:

Although schools can offer unique opportunities for structured physical activity for children, there is a tendency to cut back physical education lessons. Lithuanian scientists propose new methodology of PE. Physical education activities were taught through physical schooling, by preparing a textbook comprising two inter-related parts: a textbook, children’s notes, teachers’ book.

  1. Klizas, Š.; Klizienė, I.; Gruodytė-Račienė, R.; Buliuolienė, L.; Daukšaitė, L. Physical Education: textbook First Grade. Kaunas : Šviesa, 2016.
  2. Buliuolienė, L.; Daukšaitė, L.; Klizas, Š.; Klizienė, I.; Cibulskas, G. Physical Education: textbook Second Grade. Kaunas : Šviesa, 2017.

Response to reviewer (review 2)

What you answer at this point is not a conclusion of your study since it does not provide data on the reduction of PE sessions or that physical education sessions are the only structured activity that Lithuanian children do.

I recommend that the authors focus the introduction, discussion and conclusions in the social sphere since the objective of the study has social and non-educational implications, although the intervention is carried out in the educational system.

Author Response

Firstly, the authors want to thank your contribution to our paper. We really appreciate you taking the time out to share your experience with us.

Childhood obesity is a global problem and is it a pandemic problem, we agree. However, the fact that physical education (PE) is the only structured activity that children have access to for physical activity is not a strong argument to justify this type of program in the school setting. This refers to a structural problem of a social nature since the system is not capable of generating time-spaces for the practice of physical activity for children and adolescents. But it is not an educational problem.

Yes, we agree with you, that the fact that physical education (PE) is the only structured activity that children have access to for physical activity is not a strong argument to justify this type of program in the school setting.

The intervention had to be relevant to children health practice (focused on health promotion activities), implemented in a school setting and aimed at increasing physical activity, included all school‐attending children, and be implemented for a minimum of 8 months. In addition, the experiment was limited to randomized controlled trials and those that reported on outcomes for children. Primary outcomes included: rates of moderate to vigorous physical activity during the school day, time engaged in moderate to vigorous physical activity during the school day.

The program that you present at least from the educational point of view since there is no intrinsic didactic objective to the subject itself. In this sense, you address a social problem where the objective is also social. You could talk about linking the school with the social context, which would give more power to your work, but it is not appropriate to link it with PE subject. Therefore, in the previous review I indicated that you confuse the concepts of the intervention program with the PE subject. Confusion that they continue to make. On the other hand, you point out that "we prepared new guidelines for innovative physical education lessons and new PE methodology", but these innovative guidelines are materialized in a textbook. The textbook is not an innovation, in fact, it is the most widely used and widespread teaching material among teachers. This is a didactic material that encourages forms of learning based on repetition and uniformity. In addition, they indicate that there is an advanced version of the textbook. Does that mean that there is a non-advanced version? What is the difference between the two versions? Do you commercially sell an advanced version?

We used an advanced version textbook ant it is not commercially sell.

The physical education pack allow for a "natural" type of integration and active learning, building awareness, encouraging a sensitivity to nature and supporting healthy styles of living.

The teacher's manual has a unified structure which makes it easy to use. The recommendations and advice are straightforward. The digital version assists teachers in their planning and implementation activities.

The material seriously takes account of intercultural awareness and sensitivity. Gender representation is balanced; the two characters featured in the textbook support this approach.

 Finally, and regarding this point, in the first review I asked you a question about why you indicated that physical education lessons had a negative perception due to competition, comparison and "natural" selection that you indicated. I was concerned about the deideologization and depoliticization of physical activity even though the activities were football, basketball, gymnastics and athletics, which are fundamentally competitive, comparative and require innate biological aspects to be more successful. I need you to answer this question

Response:

We do not seek the deideologization and depoliticization of physical activity. We therefore agree with your comment and we have deleted the following sentences from Introduction:“ Although physical education lessons are often seen as funny and enjoyable, they may also trigger negative feelings, such as anxiety, due to their comparative, competitive, and evaluative nature [21]. Anxiety in physical education classes can be manifested through cognitive (e.g., negative thoughts), bodily (e.g., alteration in muscle tension), and information processing (e.g., worry and attention disruption) symptoms [21].

Response to reviewer (review 2)

I do not understand this answer because of the didactic implications it has. If the novelty is that the PE curriculum is developed, what did the children of second grade do before applying this program? Nothing? Wasn't there a curriculum that the teachers developed? I am struck by this regarding the first sentence of your response to my first review as you claim that the PE curriculum has been developing for some years.

Thank you for your important remark. In this study we test only first and second grade. We decided to choose two grades, like age indicator. It is very important measurement, on the next study we will fink about longitudinal experiments. 

Response to reviewer (review 2)

What you answer at this point is not a conclusion of your study since it does not provide data on the reduction of PE sessions or that physical education sessions are the only structured activity that Lithuanian children do. I recommend that the authors focus the introduction, discussion and conclusions in the social sphere since the objective of the study has social and non-educational implications, although the intervention is carried out in the educational system.

Response: We took into consideration your recommendations and corrected the conclusions: “Low physical activity in children is a major societal problem. The growing number of children with obesity is a concern for doctors and scientists. The focus of our study was to improve emotional well-being and physical activity in children. Since elementary school children spend most of their day at school, physical education lesson is a great tool to increase physical activity. A balanced and adapted physical education lesson can help to draw children’s attention to the health benefits of physical activity. It was established that the properly constructed and purposefully applied 8-month Physical Education Program had an impact on the Physical Activity and Emotional Well-being of primary school children (i.e., 6–7- and 8–9-year-olds) in three main dimensions: somatic anxiety, personality anxiety, and social anxiety. Our findings suggest that the 8-month Physical Education Program intervention is effective for increasing levels of physical activity. Changes in these activities may require more intensive behavioural interventions in children, or upstream interventions at the family and societal level, as well as at the school environment level. These findings have relevance for researchers, policy-makers, public health practitioners, and doctors who are in-volved in health promotion, policy making, and commissioning services.”

We also corrected the discussion: “We all have an important role to play in increasing children’s physical activity. Schools must promote and influence healthy environment for children. Most primary school children spend an average of 6-7 hours a day at school, which is most of their day time. A balanced and adapted physical education lesson provides cognitive content and training for developing motor skills and knowledge in the field of physical activity. Our 8-month Physical Education Program can give children the opportunity to increase physical activity and improve emotional well-being, which can encourage children to be physically active throughout life.”
